# OTT Streaming Distribution Strategies for Dance Performances in the Post-COVID-19 Age: A Modified Importance-Performance Analysis

**DOI:** 10.3390/ijerph19010327

**Published:** 2021-12-29

**Authors:** Jian Kim, Eunhye Kim, Aeryung Hong

**Affiliations:** 1College of Sports and Dance, Sangmyung University, Seoul 03016, Korea; artsedu@smu.ac.kr; 2Global Research Institute for Arts and Culture Education, Sangmyung University, Seoul 03016, Korea; gace.eunhyekim@smu.ac.kr

**Keywords:** dance online contents, distribution strategies, visualizing performing arts, OTT streaming, modified IPA

## Abstract

The purpose of this study was to explore strategies for distributing online content of dance post COVID-19 in Korea. And specially to discuss the distribution strategies of online performances through videoization of dance performances and OTT (over-the-top) streaming: (1) Methods: For this purpose, a survey was conducted on the distribution strategy of dance online contents for a total of 100 practitioners such as dance field, video contents, and art management. A total 91 sample were used except for defective questionnaires, and Vavra (1997)’s modified important performance analysis was conducted; (2) Results: The results of the matrix through the modified IPA analysis are as follows: first, the first quadrant included ‘quality of dance performance’, ‘platform for OTT streaming’, and ‘promotion for potential audience development’. This means that both explicit and intrinsic importance are high, and it is an important execution factor that has a positive effect on the satisfaction of the online contents of dance only if it is met. Second, the second quadrant included ‘brand awareness of choreographer or dance company’, ‘creative composition and choreography’, and ‘fee and price criteria’. This is a case of low explicit importance but high intrinsic importance, and these factors are attractive attributes that affect the satisfaction of dance online contents, although consumers do not expect it to be important. Third, the third quadrant included ‘new formats and curation’, ‘convergence technology (AR, VR, 3D, etc.) for the field sense’, and ‘online audience service (communication, membership, etc.)’. This means that both explicit and intrinsic importance are low, and if these factors are met, it can have a positive effect on the satisfaction of viewing of dance online contents. However, it does not have a negative effect even if it is not met. Fourth, in the fourth quadrant, ‘production and editing competency’, ‘quality of videos and sounds’, ‘copyright of performance creation’, and ‘fandom and audience management’ was included. This is an essential attribute in the distribution strategy of dance online contents because it has high explicit importance and low intrinsic importance, and it can have a negative impact on satisfaction when these factors are not met.

## 1. Introduction

The COVID-19 pandemic has greatly affected the world, and the arts industry as a whole is not exempt from its effects. In particular, the performance industry, which has historically been an artistic avenue that requires face-to-face gatherings, has been forced to cancel multiple series of performances or events due to concerns over the spread of infection. Due to this prolonged situation, the performing arts industry, which had been vulnerable to industrialization even before the pandemic, is facing an existential crisis, and the reorganization of its industrial direction is inevitable [1]. In the case of the dance industry, which has long been agonizing over its pure audience share, it is urgent to make a new breakthrough to attract audiences [2]. In response, the approach chosen by the performing arts industry, including the dance industry, was the launching of the online streaming platform.

There are several reasons why the performing arts industry, which has been emphasizing a sense of realism the modern era, is considering entering the online streaming platform. First, the online streaming market has continued to grow due to the emergence of smart devices, such as mobile phones and tablet PCs, since before the pandemic, and the rapid growth of data transmission has made it easier to provide content through streaming methods. This means that the market size for online content and streaming services is expanding due to the development of innovative digital technology [3,4]. Second, based on content streaming services, the accessibility of the Internet of Things (IoT), augmented reality (AR), and virtual reality (VR) markets is expected to increase in the future. In particular, the participation of major global companies, such as Netflix, Disney, and YouTube, in the information and communications technology (ICT) content streaming platform market has led to a rapid growth in related markets [5,6,7,8]. Third, demand for streaming services will surge as content platforms that are available without restrictions based on time and space become even more widely distributed. In addition, demand for over-the-top (OTT) streaming is expected to continue to increase as there are opportunities to overcome physical constraints and enjoy performance content that suits individual tastes [3].

Online streaming comes with an important basic premise. While it is clear that online streaming serves as an alternative to overcoming physical constraints, ultimately, the most important factor that determines audience’s attitude while viewing is the quality of dance performance videos. Therefore, to successfully attract audiences, the visualization of dance content must be excellent. Given this, recently, many dance companies and theaters have been strategizing the planning and curation of a digital theater for the high-quality visualization of dance. The filming and recording of performances had previously centered around the theaters, which has resulted in the production of primarily recorded videos of performances provided through online streaming. Thus, previously, the filming and recording of performances had been discussed as alternative or complementary to attending physical performances [9,10,11,12]. The US Metropolitan Opera introduced the “The Met: Live in HD” series in 2006. This series is also known to screen dance performances [13,14]. In Korea, shooting video footage of performances were first attempted through the “SAC on Screen”, which began in 2013 and centers around the Seoul Art Center under the Ministry of Culture, Sports and Tourism. Since then, Korea has provided free performance videos for the purpose of expanding the right to enjoy the performing arts and to narrow the cultural gap from the perspective of cultural welfare [15]. In the case of dance, public organizations, such as the National Dance Company, the National Ballet Company, and the National Contemporary Dance Company, have provided access to popular performances in the past for free on various online streaming media platforms. Filmed dance performances have also been provided at local cultural centers, cinemas, and libraries to allow the average citizen to experience performance culture [16]. Most of the video clips of performances in Korea were created more so as an archiving service for public welfare than for the industrialization of OTT or online streaming content.

However, the performing arts industry recently stressed the need to expand distribution platforms beyond the public interest level from an industrial point of view [17,18,19,20]. In this context, it is necessary to pay attention to the paid-for strategy that combines the filming of performances and online streaming for platform businesses. An initial example of a company introducing a paid online streaming method is the Digital Concert Hall launched by the Berlin Philharmonic in 2008. As of 2019, the company has secured 1 million registrants and 35,000 paid members, and the profits generated from this platform have been allocated back into the creation of content [21,22,23]. In the case of the UK’s digital theater, about 50 major institutions, including the Royal Opera House, the Barbie concierge, and the Royal Shakespeare Company, have joined together and operate as a singular paid OTT platform that specializes in providing videos [24]. The platform provides not only performance videos but also art education videos called “Digital Theatre Plus” to show various content models that are linked to promotion and education and lead to the development of potential and long-term audiences.

Recently, Korea’s performing arts industry has begun to accept the monetization strategy for online streaming services amid the situation of not being able to perform on-site due to the COVID-19 pandemic [25,26,27,28]. The CoM + On (CoMPAS Online) platform launched by LG Arts Center, which has been providing free online performance content during COVID-19, was the first to attempt to convert to paid services for performances in Korea [29]. This change was expected to alleviate the problems of temporal and spatial constraints, such as the one-time, sparseness, low field ability, and the resulting cost disease characteristics in performing arts. From this point of view, this study focuses on creating effective ways to attract a wider range of dance audiences through various media and distribution channels, including theaters, cinemas, internet protocol television (IPTV), and online streaming beyond attempts at alternative performance visualization. Therefore, the purpose of this study is to identify distribution strategies for online dance performances after the onset of the COVID-19 pandemic through filming and OTT streaming. Ultimately, this study seeks to provide meaningful implications to the performing arts fields and academia by overcoming the chronic problems in dance performances, such as the gap between artistry and popularity and the limitations of fundraising and industrialization.

## 2. Methods

### 2.1. Subject

This study surveyed practitioners with more than one year of work experience in the fields of dance and performing arts to determine these experts’ distribution strategies for online dance content. The survey was conducted on 100 subject samples, and 91 of them were ultimately used for analysis, as nine of them were deemed to be insincere due to the omission of questionnaire responses. The demographic characteristics of the participants are shown in Table 1.

### 2.2. Questionnaires

In this study, IPA questionnaires were used to identify the distribution strategy factors for online dance content. To this end, factors and questionnaires were constructed based on prior studies consistent with this research, which considered relevant prior studies in various fields, including plays, classical music, and dance, which predicted the future of non-face-to-face performance arts. Then, Delphi surveys were conducted on expert panels to verify the validity of the factors regarding the distribution strategy of online dance content. The Delphi panel consisted of nine experts in dance and performing arts. To ensure content validity, the study conducted a modified Delphi technique. Consequently, 13 final factors were derived, as shown in Table 2, by revising and supplementing the initial questions by reflecting upon the common opinions of the expert panel. These 13 factors are (1) quality of dance performance, (2) brand awareness of choreographer or dance company, (3) production and editing competency, (4) quality of videos and sounds, (5) new formats and curation, (6) convergence technology for field sense, (7) creative composition and choreography, (8) services for online audiences, (9) copyright of performance creation, (10) fee and price criteria, (11) platform for OTT streaming, (12) fandom and audience management and (13) promotion for potential audience development. The questionnaire consisted of a total of 34 items, including 13 items gauging importance, 13 items for opinion regarding performance, 1 question for overall satisfaction, and 7 items for demographic characteristics. A five-point Likert scale (1 = not at all, 5 = very much) was used as the survey tool to gather responses.

### 2.3. Data Analysis Modified Importance-Performance Analysis

The importance-performance analysis as outlined by Martilla and James (1977) has been useful for the comparative analysis of attributes in quadrants based on the importance of the *X*-axis and the satisfaction of the *Y*-axis [30]. However, some have raised the point that the limitation of these traditional IPAs does not meet the basic assumption that they should be linear and symmetric. As such, a new method called the Modified Importance-Performance Analysis was suggested [31,32,33]. In this regard, Vavra (1997) proposed a modified IPA technique based on the Three-Factor Theory presented in a study by Noriaki, Seraku, Takahasi, and Tauji (1984) [33,34]. The Three-Factor Theory compensates for the asymmetric problem of traditional IPAs. Unlike traditional IPAs, this theory describes the importance of respondents by directly evaluating for “Explicit Importance” and the standard regression coefficient for this is also referred to as “Implicit Importance” [33]. As such, the intrinsic importance derived from statistics is more reasonable for predictions than explicit importance measured directly, and it has been reported that it better explains consumers’ inherent psychology. Thus, this study applied modified IPA techniques using the Vavra (1997) regression coefficients to simultaneously compare and analyze the attributes of the revenue strategies of online dance content streaming initiatives on a quadrant of a matrix and derive a comprehensive strategy method [33].

This study conducted a survey of self-assessment techniques online from 14–23 September 2020. Of the 100 samples, only 91 responses were used for analysis, and nine unreliable ones were excluded. We utilized SPSS version 25.0 (IBM, New York, NY, United States) for frequency analysis, multiple regression analysis, and modified IPA. This study presented the mean of each property derived in a quadrant of a two-dimensional matrix based on the *X*-axis (explicit importance) and the *Y*-axis (intrinsic importance). Specifically, the “Important Performance Factor” where the demand for online dance content distribution strategies and satisfaction with the quality of services were generally proportional and the “Excitation Factor”, which increases satisfaction as the service needs are met, were presented in quadrants 1 and 2, respectively. In addition, the “Unimportant Performance Factor” consisting of factors with relatively low intrinsic and explicit importance to the online dance content distribution strategy and the “Basic Factor” that can lead to dissatisfaction if service needs are not met were presented in quadrants 3 and 4 respectively.

## 3. Results

### 3.1. Importance of Distribution Strategy for OTT Streaming of Dance

The results of the importance analysis on OTT streaming distribution strategies for dance performance showed that “copyright of performance creation (4.47)”, “production and editing competency (4.46)”, “quality of dance performance (4.40)”, “promotion for potential audience development (4.38)”, “quality of videos and sounds (4.29)”, “platform for OTT streaming (4.22)”, “fandom and audience management (4.22)”, “brand awareness of choreographer or dance company (4.16)”, “services for online audiences (4.16)”, “new formats and curation (4.10)”, “convergence technology for field sense (4.03)”, and “fee and price criteria (3.88)”. As such, the “copyright of a performance creation” was shown to have the highest importance ranking, which could be interpreted as reflecting the perception for the need for copyright protection, which has recently become an issue. It was also confirmed that the online distribution of dance content’s quality improvement related to the professional competencies of dance performance videos and promotional strategies, which are necessary to attract more audiences, should be promoted. Although the lowest ranking “fee and price criteria” condition was not directly related to the production of online dance content, it was identified as a factor that needs to be continuously improved in the long-term.

### 3.2. Satisfaction with the Distribution Strategy for the OTT Streaming of Dance

As a result of the satisfaction analysis on the OTT streaming distribution strategy of dance performances showed that “quality of videos and sounds (3.79)”, “quality of dance performances (3.67)”, “production and editing competency (3.62)”, “copyright of a performance creation (3.49)”, “brand awareness of choreographer or dance company (3.47)”, “new formats and curation (3.47)”, “platform for OTT streaming (3.45)”, “creative composition and choreography (3.38)”, “services for online audiences (3.32)”, “convergence technology for field sense (3.27)”, “fee and price criteria (3.23)”, “promotion for potential audience development (3.22)”, and “fandom and audience management (3.19)”. High satisfaction levels, such as with “quality of video and sounds”, “quality of dance performance” and “production and editing competency” have shown that overall satisfaction with the level of technology in dance video content production has been achieved. However, satisfaction with “promotion for potential audio development” and “fandom and audio management” was relatively low. It can be interpreted that an active promotion and management system should be put in place for audience development. In particular, these factors have confirmed that promotional services for audience development should be continuously provided as they are likely to directly affect audience satisfaction.

### 3.3. Matrix Results regarding Distribution Strategy for the OTT Streaming of Dance

The results of the matrix from the modified IPA analysis are shown in Table 2 and Figure 1. Explicit importance, as shown on the *X*-axis, was shown as having a minimum value of 3.8 and a maximum value of 4.5 based on the mean value of 4.2. The intrinsic importance of the *Y*-axis was shown as a minimum value of −0.4 and a maximum value of 0.5 based on the mean value of 0.06. The total number and quadrant of each variable are as shown in Figure 1. First, “quality of dance performance”, “platform for OTT streaming” and “promotion for potential audience development” were presented in quadrant 1 as representing important execution elements. These factors are of both high explicit and intrinsic importance and are important enforcement factors that must be met to positively affect the viewing satisfaction with online dance content. Second, “brand awareness of choreographer or dance company”, “services for online audiences” and “fee and price criteria” were presented in quadrant 2, indicating they are attractive elements. These factors are of low explicit importance but high intrinsic importance, and these factors are attractive attributes that affect the viewing satisfaction with online dance content, although consumers do not expect much. Third, “new formats and curation”, “convergence technology for field sense” and “Services for online audiences” were included in quadrant 3 representing the unimportant elements of execution. This is both low in explicit importance and inherent importance, and if these factors are met, they may have a positive effect on viewing satisfaction with online dance content, but if not, they do not have a significant negative impact. Fourth, “production and editing competency”, “quality of videos and sounds”, “copyright of performance creation” and “fandom and audience management” were included in the quadrant representing essential elements. These factors are of high explicit importance and low intrinsic importance, meaning that if these factors are not met, this can have a significant negative impact on satisfaction, which is an essential attribute in the distribution strategy of online dance content.

## 4. Discussion

The purpose of this study was to discuss the distribution strategy of online performances through the visualization of post-COVID-19 dance performances and OTT streaming; the following several major points of discussion were suggested. This may be a turning point in other industries such as the sports industry, which has been considering the influence of media and rights to relay due to the flow of capital and industrialization, and the entertainment industry, which is aggressively investing in high technology and capital to overcome physical constraints.

The first quadrant is an important execution area with satisfaction and dissatisfaction characteristics that consumers can experience, including “copyright of performance creation”, “production and editing competency”, “quality of dance performance” and “promotion for potential audience development”. Factors in the first quadrant could have a positive impact on consumer satisfaction. Although direct comparison with this study is not possible, it can be inferred from a service quality dimensional model that consists of three factors: service product, service delivery, and service environment as proposed by Rust and Oliver (1994) [35]. It was found that important factors in the distribution of dance performance online content to audiences were also required in the general quality of service model. On this basis, it can be interpreted that the level and method of performance delivered to audiences regardless of the type of performance (online or offline) is important. Live-streamed content is accessible in that it can be viewed using smart devices without space or time limitations, unlike conventional live performance broadcasts. In addition, it enables free appreciation regardless of the viewing manner or etiquette of performance halls and it relieves the inconvenience of sitting in a fixed seat to appreciate it. Considering this, it could be predicted that a new value chain will be formed, moving away from the value chain connected by existing creation, production, distribution, and consumption [18,36,37,38]. A substructure of an infrastructure should be created to enable the production, distribution, and consumption of performing arts online centered on existing performance halls. Therefore, online dance performances are expected to form a new value chain structure that leads to suppliers such as performance halls and performing arts organizations, content promotion polices that support the performance visualization base, platforms that distribute the services of digital theaters, and paid audiences.

The second quadrant includes “brand awareness of choreographer or dance company”, “services for online audiences” and “fee and price criteria” as unexpected areas in which consumers could experience satisfaction. Generally, when elements belonging to the second quadrant are satisfied, this has a positive effect on consumer satisfaction; however, they do not have a negative effect on satisfaction even if they remain unsatisfied. It is necessary to pay attention to the results of this study. In recent years, the global media market has undergone a digital revolution due to the emergence of new technologies such as digital TV, OTT streaming services, and online mobile games. This digital revolution has created a major shift in the overall aspects of the industry through new approaches that were unimaginable in the analog era [39]. The fundamental reason audiences consume online dance content is that the online environment has been activated by COVID-19. In addition, Chesbrough (2007) suggested that the functions of the business model are activities such as value creation, price setting, and technology upgrades for customers according to service provision [40]. What is noteworthy here is that the key to paying for online performances is that a standard price that is differentiated from the existing on-site performances is required. This is because experts forecast that it will be difficult to mobilize free audiences, let alone paid audiences, with most of the existing online performance methods [24]. In addition, the streaming platforms of portal sites and mobile telecommunication service providers, which can be the point of contact for online relay, must also take into account the reality that advertising fees rather than usage fees contribute to the main revenue [14]. According to this, brand awareness of dance performances, online attractiveness, and paid prices are not yet important but can be interpreted as areas considered important in the future.

The third quadrant includes “new formats and curation”, “convergence technology for field sense” and “services for online audiences” as areas that have a positive effect on consumer satisfaction when satisfied but do not have a negative effect if they go unsatisfied. The convergence of technology for the presence and the attributes of online audience services is noteworthy in this result. Currently, media technology that actively converges in the performing arts field can be classified into four areas: virtual reality (VR), holographic performance, real-time interactive motion sensor recognition and graphics grafting technology, and expansion of stage video media technology [39,41]. However, it is essential to use expensive equipment to appreciate content by reflecting performance convergence technology, and a difference in content immersion can be experienced according to the specifications. In addition, the implementation of stage actors’ movements using high-performance capture technology, real-time motion sensor recognition, and graphic technology attempts to integrate traditional performing arts and technology areas, but the technology to reflect all movements of actors has not yet been equipped. Due to this weakness, content that reflects convergence technology is generally produced in segments that last 5–10 min or less rather than the total performance [41,42,43]. In addition, it should be considered that the existing online content used as tools for new distribution included in the third quadrant is proportional to the satisfaction of consumers and the satisfaction of the service requirements, but the service is low priority [44,45,46]. Therefore, excessive efforts or cost input should be strategically selected Specifically, it seemed to have succeeded in creating an audience in the early stages of performance visualization, but the effects of quantitative and qualitative dimensions have not been clearly revealed, and it is experiencing difficulties in continuous growth [17,18]. Unlike overseas markets, where the imaging business has already been established in earnest, in the case of Korean dance and classical genres, the Seoul Arts Center’s SAC on Screen project is the only one of its kind. On the other hand, the entertainment industry, which offers online-only paid concerts, uses technologies such as AR, VR, and 3D to provide attractions that cannot be accessed offline [47]. In particular, in the K-POP field, due to the use of 5G services, attention is paid to cultural technology that combines advanced technologies such as augmented reality, artificial intelligence, and big data [48]. In this way, it is necessary to devise a way to improve the quality of the imagery of dance performances and to strengthen industrial competitiveness through the preemptive entertainment industry imaging business and distribution strategy.

The fourth quadrant is an essential area with attributes of high explicit importance and low intrinsic importance, including “production and editing competency”, “video and sound quality”, “copyright of performance creation” and “fandom and audience management”. What is noteworthy in this result is that performances include essential elements in the delivery to consumers. In classic live streaming, the video content and physical environment qualities are in line with the results of Kim and Lim (2020), which verified the relationship between consumer satisfaction and viewing [25]. In fact, New York Met Opera Live, which started relaying live performances for the first time recently, is known to edit and transmit videos taken with up to 10 cameras along with the recruitment of film directors. This is to include performances in a realistic manner, and it is expected that technical support for platforms that transmit performances and capabilities for video and sound technology will be required continuously. In addition, the essential factor in streaming online dance content is the copyright of performance creation. Online streaming services are designed to protect the copyrights of works, but there are many cases of abuse [5,49]. In this regard, copyright protection technology for online streaming services has been proposed. Recently, the number of copyright dispute cases has increased with the industrialization of performing arts and pure creative performances in Korea. In Korea, dance and silent drama are defined as types of theatrical work because they express ideas and emotions through gestures or movements as in a play rather than the means of language. However, countries such as Germany, Japan, France, and the United States define dance and silent works as independent works. In this regard, it suggests that, despite the existence of separate creative concepts and expression methods for each art genre, the Korean dance industry should be recognized as an independent concept in opposition to the classification of dance works as a schedule of theatrical works without considering them [50], Currently, dance could be divided into choreography and performances (dance) that perform the choreography, which distinguishes between the rights of choreographers and performers under the Copyright Act. In addition, the copyright of the creation should be protected by reflecting the rights of the planning producers, creators, performers, agencies, productions, etc. in the planning, performing, and distribution stages, since dances have various stages, Thus, regulations related to dance performance should be revised in advance, and efforts should be made to ensure the rights of creators and performers through the standard contract for online performance and distribution and to establish a fair environment for the use of performance videos.

Based on these points of discussion, the possibility of a new market for dance industrialization can be raised with respect to the fact that the imaging of dance is being popularized in a similar form to other video genres such as advertisements and movies. It can also be linked to new publicity and management systems such as viewing services to attract online audiences, teaser trailers, lecture performances, and dance films and create convergence with popular art. First, we can take as an example the influence of dance and dance films in advertisements. Apple’s earphones and GAP’s apparel advertisements feature scenes that pay homage to Yoann Bourgeois’ dance film The Great Ghost [51]. In addition, the trend of combining dance with popular art and other video genres in various forms can provide an opportunity for the popularization of dance. Thom Yorke, a vocalist of the rock band Radiohead, produced an art film Anima, which contained contemporary dance performances, alongside film director Paul Thomas Anderson and choreographer Damien Jalet through Netflix in 2019 [52]. The Korean idol group BTS became a sensation with their traditional Korean dance performance at the 2018 Melon Music Awards and when they released an art film in collaboration with MN Dance Company, a modern dance company, with their music video for “Black Swan” [53]. As such, the visualization of dance oriented toward popularity has the potential to influence various industries, as attractive elements that lead the market trends are latent.

## 5. Conclusions

This study yielded the following interpretations based on the importance and implementation of distribution strategies for the online streaming of dance performances. To this end, a survey was conducted on the distribution strategies of online dance content targeting 91 practitioners working in the fields of dance, video content, performing arts, and art management, and a revised IPA analysis was conducted. The result is as follows. First, while “copyright of performance creation”, “production and editing competency” and “quality of dance performance” were factors of high importance, the importance of “convergence technology for field sense” and “services for online audiences” was considered to be low. These results show that the improvement of copyright and video content quality for online dance content should come first. Second, satisfaction with technologies is essential for the production of dance video content when there is frequent distribution of online dance content, but the importance of satisfaction with factors related to audience development is low. These results confirm that a new type of promotion and management system should be involved, including viewing services to attract online audiences, teaser videos, lecture performances, dance films, and convergence with pop art. Third, revised IPA results used to derive online dance content distribution strategies showed that “quality of dance performance”, “platform for OTT streaming” and “promotion for potential audience development” were important execution factors. Based on these conclusions, this work suggests directions for the distribution strategies for online dance content as follows: First, excellent performance halls that have preemptively led the online live performance industry and active strategies for analyzing and benchmarking the K-pop entertainment industry are required. Second, environmental support for the establishment of industrial infrastructure should be provided by considering the domestic situation where video production, digital theater, and online distributors are insufficient. Third, these strategies should be carried out in a way that supports the ecosystem of industrialization by drastically revising existing cultural and artistic policies that have focused only on the promotion of the public’s cultural enjoyment. This requires the creation of a basis for imaging technology as well as the activation of policy interventions and support for the distribution structure of OTT streaming platforms.

This study is meaningful in that it empirically analyzed and presented the importance–satisfaction of online dance performance distribution attributes as basic data to help revitalize the dance performance industry amid the increasing importance of online dance distribution due to COVID-19. Based on the results presented in this study, it is necessary to proceed with more sophisticated empirical studies on the characteristics of online dance performance distribution.

## Figures and Tables

**Figure 1 ijerph-19-00327-f001:**
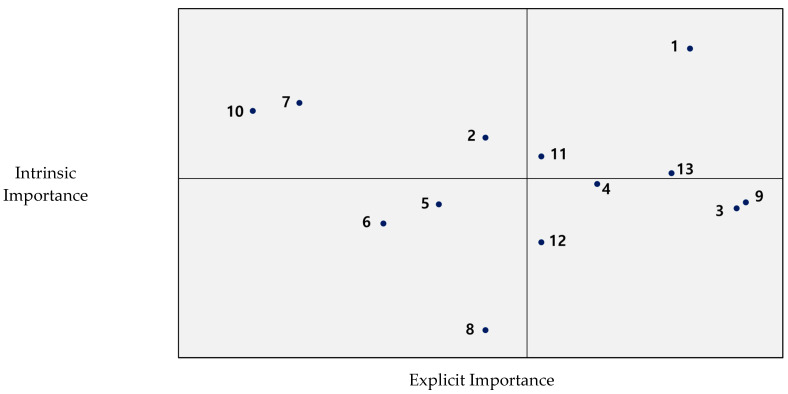
Modified IPA matrix of distribution strategy for the OTT streaming of dance.

**Table 1 ijerph-19-00327-t001:** Demographic characteristics of the study.

		Frequency (n)	Percentage (%)
Gender	Male	21	20.8
	Female	70	79.2
Ages	20s	70	91.7
	30s	16	4.2
	40s	1	1.4
	50s	4	2.8
Work Experience	More than 1 year~less than 3 years	14	15.4
	More than 3 years~less than 6 years	20	22.0
	More than 6 years~less than 10 years	30	33.0
	More than 10 years	27	29.7
Total		91	100

**Table 2 ijerph-19-00327-t002:** Results of revised IPA.

Rank	Factor	Explicit Importance	Intrinsic Importance	Quadrant
1	Quality of dance performance	4.400	0.41	Quadrant 1
2	Brand awareness of choreographer or dance company	4.155	0.17	Quadrant 2
3	Production and editing competency	4.455	−0.01	Quadrant 4
4	Quality of videos and sounds	4.288	0.04	Quadrant 4
5	New formats and curation	4.100	−0.007	Quadrant 3
6	Convergence technology for field sense	4.033	−0.059	Quadrant 3
7	Creative composition and choreography	3.933	0.269	Quadrant 2
8	Services for online audiences	4.155	−0.349	Quadrant 3
9	Copyright of performance creation	4.466	−0.002	Quadrant 4
10	Fee and price criteria	3.877	0.247	Quadrant 2
11	Platform for OTT streaming	4.222	0.012	Quadrant 1
12	Fandom and audience management	4.222	−0.11	Quadrant 4
13	Promotion for potential audience development	4.377	0.077	Quadrant 1

## Data Availability

The data are not publicly available due to privacy issues.

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
