# Peer review of "OTT Streaming Distribution Strategies for Dance Performances in the Post-COVID-19 Age: A Modified Importance-Performance Analysis"

_ijerph, 2021, doi:10.3390/ijerph19010327_

Round 1
Reviewer 1 Report
In my opinion, the article is very correct and deals with a very specific phenomenon in a very interesting way. The strategies used by OTT platforms are very significant and, the specific case of Dance Performances is relevant.
The work presents a clear object of study and a highly developed methodology. From my point of view, the main problem of the article is given by two issues. First of all, the conclusions are short, it would be necessary to expand them a little more. On the other hand, I miss a specific Discussion section.
Reviewer 2 Report
You have selected an interesting topic and approached it from an insightful perspective. You also show a great degree of transparency about your research methods, which I appreciate. There are, though, passages with some room for improvement:
When describing the subject, you state (p.3, lines 125-6) that "The survey was conducted on 100 subject samples, and 91 of them were 125 ultimately used for analysis as nine of them that were deemed to be insincere". Could you explain on what grounds you discarded this subject? Also, explaining the criteria used to recruit the 100 practitioners would increase the soundness of your methodology.
In your conclusions (p. 8, lines 347-8), you state that "These results show that the improvement of copyright and video content quality for online dance content should come first", and though it seems the reflection has interesting repercurssions, the argument might need to be further developed to be fully understood. This is a pattern that I've detected for the general conclusions as a whole: the statements are well structured, they build up on the previous Discussion rather than just repeating the data from the results sections, but quite frequently they would benefit from more detailed explanation on the links the author(s) see to reach such conclusions.
Author Response
We will upload a revised version that reflects the opinions of the reviewer.
Thanks for the good comments.
